# Environmental Factors Affecting Feather Taphonomy

**DOI:** 10.3390/biology11050703

**Published:** 2022-05-03

**Authors:** Mary Higby Schweitzer, Wenxia Zheng, Nancy Equall

**Affiliations:** 1Department of Biological Sciences, North Carolina State University, Raleigh, NC 27606, USA; wzheng2@ncsu.edu; 2North Carolina Museum of Natural Sciences, Raleigh, NC 27601, USA; 3Department of Geology, Lund University, 223 62 Lund, Sweden; 4Museum of the Rockies, Montana State University, Bozeman, MT 59717, USA; 5ICAL Facility, Montana State University, Bozeman, NC 59717, USA; equall4@msn.com

**Keywords:** feather, taphonomy, degradation, keratin, microbes, CO_2_, apatite, melanin

## Abstract

**Simple Summary:**

This study seeks to test the effect of burial/exposure, sediment type, the addition of feather-degrading microbes, and the addition of minerals on feather preservation, and for the first time, compares these states in ambient vs. elevated CO_2_ atmospheres to test the effect of CO_2_ on degradation and/or preservation under various depositional settings.

**Abstract:**

The exceptional preservation of feathers in the fossil record has led to a better understanding of both phylogeny and evolution. Here we address factors that may have contributed to the preservation of feathers in ancient organisms using experimental taphonomy. We show that the atmospheres of the Mesozoic, known to be elevated in both CO_2_ and with temperatures above present levels, may have contributed to the preservation of these soft tissues by facilitating rapid precipitation of hydroxy- or carbonate hydroxyapatite, thus outpacing natural degradative processes. Data also support that that microbial degradation was enhanced in elevated CO_2_, but mineral deposition was also enhanced, contributing to preservation by stabilizing the organic components of feathers.

## 1. Introduction

Feathers are arguably the most complex integumentary structures in the entire animal kingdom. The evolutionary origins of feathers are still debated, but growing evidence from both molecular studies in extinct theropods [1,2,3,4,5,6,7,8] and living birds (e.g., [9,10,11,12,13,14,15,16,17,18]), as well as numerous fossil discoveries of structures morphologically consistent with feathers (e.g., [4,19,20,21,22,23,24,25]) indicate that feathers arose from filamentous structures first identifed in some theropod dinosaurs and birds more than 160 million years ago (e.g., [2,26,27]). However, some data suggest that integumentary structures similar to those from which feathers derived may have been present at the base of Dinosauria [28,29] or perhaps, the base of Archosauria ([30,31] and references therein). Because modern feathers are not biomineralized in life (contra [32,33]) their persistence in the fossil record is counterintuitive, but critical. The impressions of feathers in sediments surrounding skeletal elements led to the identification of *Archaeopteryx* as the first bird [34,35], but there was no organic trace with this specimen to suggest that any original material remained. However, the first specimen attributed to *Archaeopteryx* was a single, isolated feather [36]. This specimen presented differently from feather impressions surrounding the skeletal remains, instead visualized as a carbonized trace clearly distinct from the embedding sediments, suggesting that taphonomic processes resulting in preservation differed between the isolated feather and the skeletal specimen. The environmental factors resulting in these different modes of preservation remain relatively unexplored.

For any organic remains to be preserved in deep time, they must be stabilized before they can degrade [37]. Although many taphonomic modes may result in the preservation of feathers (e.g., carbonized film [36,38], sediment impressions [34,35], three-dimensional filaments [3,39], amber preservation [40,41], bacterial mediation [42,43], or other stabilizing processes (e.g., [42,44,45,46])), few of these have been subjected to rigorous experimentation, particularly at the molecular level (but see [47]). It is likely that preservation processes differ for every fossil element and/or environment and arise from a complex interplay between the molecular composition of the original structures and the geochemical properties of the surrounding depositional matrix. Factors contributing to feather preservation are testable by approximating naturally occurring conditions in laboratory experiments.

Here, we examine multiple environmental factors that may, to varying degrees, affect the preservation of feathers in the fossil record. We discontinued the experiments after six weeks, because previous experiments (feathers degraded in sandy settings with added microbes) have shown that within this time period, the degradation of feathers was almost complete (unpublished data). We tested feathers buried vs. unburied; soaked in natural pondwater, sterile water, or pondwater after incubation with feather-degrading bacteria; the addition of solubilized hydroxyapatite (HA) to waters; and burial in sand vs. mud. Finally, we repeated the experiment in atmospheres elevated in CO_2._ Although the value we employed is higher than proposed for the entire Mesozoic, it is consistent with values proposed for the Ypresian [48]. Thus, this value is in line with what has occurred without human intervention, allowing us to test the direct effect of CO_2_ in a shorter time period. We observed that in at least one case, there was no material remaining to be tested at six weeks; degradation was complete.

## 2. Methods

We subjected the black and white feathers of an extant magpie (*Pica hudsonia*) to these environmental conditions to test their effect on preservation and/or degradation (see Table 1). These feathers allowed us to also test the hypothesis that melanized feather regions would show greater preservation than un-melanized ones [49]. See Appendix A for additional experimental details.

We hypothesized that rate, degree, and pattern of the degradation of feathers might differ in atmospheres elevated in CO_2_ relative to today’s levels, so we divided all experimental conditions into “elevated CO_2_” and “ambient” atmospheres. Within these two environments, we tested the effect of burial vs. surface exposure; native microbial populations in surface waters (pondwater) or the addition of feather-degrading microbes (*B. licheniformis*) to those microbes naturally present in pondwater; clean porous sand (No. 1113 Premium Play Sand, Quikrete) vs. natural pond sediments, which were a silty-to-clay mix; and an addition of 6 mM calcium hydroxylapatite (HA; Ca_10_(PO_4_)_6_(OH)_2_), (0.1 MCaCl_2_ solution mixed with NaH_2_PO_4_ (0.1 M) with volume ratio 10:6)) [50] to mimic the higher concentration of this mineral within pore waters that occurs under elevated CO_2_ and concurrent lower pH [51,52,53]. This would be more likely to occur in Mesozoic than present-day waters because of the increased acidification and resulting mineral solubilization brought on by elevated CO_2_ [52,53,54,55]. To test the role of microbes in mediating HA mobilization, we added this mineral to both E-Pure water and pondwater before adding to feathers. Although we used E-Pure water in the last two conditions as controls, some microbes were present, arising from the natural sediments and/or the feathers themselves.

Table 1 illustrates the conditions tested and the abbreviations used in further descriptions. Feathers were cut into sections, marked for periodic recovery, and sampled every two weeks (see Appendix A for more information). Only data from week six are shown herein, except for buried feathers with added microbes in elevated CO_2_ (CPBB). Under this condition, no material remained for testing; we show data from the four-week timepoint for this condition in all figures except SEM images for elevated CO_2_ conditions. When we could discern original color, it was noted in the figure legend.

In addition to micromorphological changes, we tested the effect of degradation on antibody binding using a polyclonal antibody raised against extant mature white feathers, which are comprised almost completely of feather corneous β-protein (CβK [39]). In all cases, there was a bacterial component to the experimental conditions, including the E-Pure water conditions, because feathers still rested on unsterilized sediments infiltrated with bacteria in pondwaters or present on the feathers themselves. This is appropriate, as there are no naturally occurring environments that are devoid of bacterial influence. Normal microbial flora would be expected to be present in waters, sediment/sand, and the feathers themselves in all conditions.

## 3. Results

### 3.1. Transmitted Light Microscopy (LM)

Figure 1 shows feathers subjected to the above conditions and subsequently imaged using LM under ambient atmosphere (normal air, at room temperature), subjected to the above conditions. Images were taken after six weeks of degradation. For each condition (row), the two left-most panels represent buried feathers, and the right two panels are feathers exposed at the surface.

Under ambient conditions, feathers showed little obvious degradation after two weeks (not shown), but at six weeks, in most cases, degradation was obvious in both buried and exposed feathers with natural pondwater, and greater when *B. licheniformis* was added. In the former (PB, PE; Figure 1A–D), white feather regions that were buried showed more extensive fraying than observed for black feathers, and the buried feathers were more degraded than those exposed at the surface. Both black and white regions of the feathers were significantly degraded in the buried condition (Figure 1A,B), but in the unburied condition, the black regions of the feather were generally less degraded than the white regions (Figure 1C,D). In some regions of the white buried feather, no barbs could be seen. The white regions of the rachis were crumbling and covered in a fine, crystalline material. When *B. licheniformis* was added to the feather setup (Figure 1E–H), degradation was increased over the pondwater-only condition. In general, degradation, as measured by fraying, absence of barbs, and loss of integrity, was greater when feathers were buried, and visual inspection showed that white feather regions suffered greater degradation than did black ones, although white rachises were relatively intact.

When HA was added to pondwaters, preservation was enhanced in both buried (PMB) and unburied (PME) states (Figure 1I–L). In all cases, the barbs and barbules were intact and color was still discernible. Similarly, there was virtually no degradation visible in feathers exposed to E-Pure water with added HA (ESMB/E), even after six weeks (Figure 1M–P). The barbs, barbules, and vane structures were intact in both the white and black regions, both buried (M,N) and unburied (O,P), despite the presence of native microbes in the sediments and water. This was also seen in the feathers incubated with E-Pure water without added minerals (ESB/E, Figure 1Q–T), although some fraying of the white barbs was visible in the exposed regions (Figure 1T).

Figure 2 shows an experimental set-up identical to Figure 1, but degradation was conducted in an elevated CO_2_ atmosphere (5000 ppm), reflecting estimates of some of the highest levels of naturally occurring (i.e., non-anthropogenic) atmospheric CO_2_ of the Phanerozoic (e.g., [48]). In each row of images taken after six weeks (except panel E,F, at four weeks), the first two panels were buried and the second two exposed on the surface, as above.

Generally, under light microscopy, degradation appeared more intense in the elevated CO_2_ atmosphere than ambient in CP or CPB conditions. Buried feathers were degraded to completion in CPBB and CPBE, possibly indicating an upregulation of urease or carbonic anhydrase enzyme production by these microbes under elevated CO_2_ [56]. Because no feathers were recovered at the six-week endpoint, here we include the four-week buried data point (Figure 2E,F). In all other cases, both black and white regions of the feather could still be differentiated under transmitted light; barbs and barbules appeared frayed by the sixth week but were still intact. Buried feathers appeared slightly more degraded than unburied ones, as seen in ambient atmospheres as well, and black regions of feathers in CPB/CPE (Figure 2A–D) and CPBB/CPBE (Figure 2E,H) were slightly less degraded than white feathers.

### 3.2. Scanning Electron Microscopy (SEM)

We used SEM to test the hypothesis that black feathers are more resistant than white feathers under the conditions described above. The SEM images in Figure 3 and Figure 4 were taken after six weeks of degradation. The conditions under which data were collected followed what is shown in Figure 1 and Figure 2. Feathers were differentiated according to color, when possible to discern. Figure 3 represents degradation in ambient atmosphere; Figure 4 is feathers degraded in elevated CO_2_ atmospheres.

In ambient atmosphere, feathers degraded for six weeks in pondwater showed little difference between buried (PB, Figure 3A–D) and exposed (PE, Figure 3E–H). Highly aligned, confluent microbial mats (m) could be seen on the surface of the fragmented keratinous outer cover in both black (3A–B) and white (3C–D) feathers in both PB and PE samples. Fungal hyphae (F) could also be seen. Occasionally, mineral crystals (Figure 3A, arrowheads) could be seen.

When feathers were exposed on the surface (Figure 3E–H), degradation appeared to be slightly less, but confluent, ordered microbial bodies (m) were seen on the surface of the feathers in all cases. The keratinous outer sheaths were less fragmented, but invasive microbial populations were still visible. Microbial impressions, or “voids” (V), were visible on the surface of the keratinous sheath (Figure 3F,G). Figure 3H shows a bundle of fraying keratin fibers (kf). Mineral crystals and/or diatoms were also visible (Figure 3G, arrowheads). The keratin sheath was densely pitted (Figure 3F, left).

When feathers were soaked in pure cultures of *B. licheniformis* and then added to the normal pondwater flora, degradation was greatly intensified in both the buried (PBB, I–L) and unburied (PBE, M–P) feathers. The keratinous material from the buried samples was fragmented and riddled with holes. Microbial bodies (m) were visible in all panels. In both PBB and PBE, degradation followed a different pattern, clearly visible as microbial tunneling (arrows, Figure 3M–P). This was not observed unless *B. licheniformis* were added, and may be specific to this microbial group. Microbial bodies were not as readily visible in the exposed feathers relative to the buried samples. Bundles of keratin fibers (kf) were loose and ropy. Small regions of apparent crystal growth (arrowheads) can be seen in in direct association with the microbes in Figure 3I.

Buried feathers in natural pondwaters to which HA had been added (Figure 3Q–T) showed better preservation than previous conditions. Barbs (B) and barbules (BL) were still visible (Figure 3Q) and accumulations of geometric crystal growth (arrowheads) were seen on all feather surfaces, both black (Figure 3Q,R) and white (Figure 3S,T). In some cases, an amorphous film could also be seen coating feather surfaces (Figure 3S,Y,Z, (#)). However, deep in the crystal growth and amorphous film, fibers of keratin were still aligned and appeared intact. The unburied feathers with added HA (Figure 3U–X) also showed better preservation than the previous conditions. Feather barbs (B) and barbules (BL) were visible, and showed virtually no degradation; they appeared to be coated in mineral crystals, which may have stabilized the organic structures [57,58,59]. Fungal hyphae could be seen (Figure 3V,W (F)) but microbial bodies were rare in this condition. Figure 3X shows a fibrous mat of material (fm), the identity of which is uncertain. It was not possible to differentiate black and white feathers.

Buried feathers incubated with E-Pure water with added HA (ESMB) under ambient conditions showed virtually no degradation (Figure 3Y–BB). In some regions of these feathers, fungal hyphae were prevalent (Figure 3Y, (f)), and the same amorphous deposits (#) and mineral crystals (arrowheads) were visible on the feather surfaces. In Figure 3Z, this amorphous and slightly crystalline film (#) appeared to completely coat regions of the buried feather. Barbs, barbules, and hooklets (hl) were visible (Figure 3AA). Degradation did not differ measurably between black (Figure 3Y–Z) and white (Figure 3AA,BB) feathers.

Feathers exposed on the surface and incubated with E-Pure water and added HA (ESME) showed exceptional preservation, with virtually no degradation, although mineral deposition was noted on the surfaces of some regions. There was no obvious difference in preservation between black (Figure 3CC,DD) and white (Figure 3EE,FF) feathers. Unaltered barbs and barbules (BL) could be seen, and filaments of keratin comprising these were visible as aligned fibers (KF). Flattened fungal hyphae (F) were visible, in some cases with outgrowths of bushy, finely crystalline structures (Figure 3DD, (Cr)). These “floret-like” structures were also visible in other regions of unburied feathers and were associated with fungal hyphae. Very few microbial bodies (m) were observed in either buried or unburied feathers in this condition, but a few could be seen in association with this floret-like structure, and they appeared both as bacilliform and coccoid structures (Figure 3FF, (m)).

Feather preservation was also greater when incubated in E-Pure water in both buried (ESB, Figure 3GG–JJ) and unburied (ESE, Figure 3KK–NN) states, consistent with LM data, and virtually no differences were seen between black (Figure 3GG,HH) and white (Figure 3II,JJ) regions. Fungal hyphae (F) were prevalent in buried feathers (and appeared, in some cases, to be associated with a thin amorphous coating tentatively identified as biofilm (bf)) (Figure 3HH,II). This amorphous film appeared to coat buried feather barbs almost completely in some regions.

Some regions of amorphous film (bf) were associated with fungal hyphae in the unburied feathers (Figure 3KK–NN) as well. Feather barbs and barbules were clearly visible, and although crystals had deposited on the surface (arrowheads), the structure of the feather was intact. Even though no excess HA was added to these burial conditions, in some areas, mineral crystals were still visible. The fibrous texture of the keratin comprising the barbs was intact (Figure 3NN).

Figure 4 shows feathers exposed to the same conditions as Figure 3, but conducted in an elevated CO_2_ atmosphere. When feathers were buried and incubated with natural pondwaters (CPB, Figure 4A–D), virtually no feather structure remained after six weeks, but fungal hyphae were evident (F) and the feathers revealed highly degraded, pitted surfaces (e.g., Figure 4C,D) colonized by fungi (F). Needle-like crystals (arrowheads) were observed on the surface (Figure 4A,B). The original color of the feathers could not be discerned.

The unburied feathers in this elevated CO_2_ pondwater environment (CPE) fared only slightly better (Figure 4E–H). Some of the barbules (Figure 4E–F, (BL)) remained intact, but in parts they were covered with a regional overgrowth of fibrous material (fm). Mineral crystals (arrowheads) and occasionally diatoms (dt) could be seen. However, in other regions, the surface was riddled with holes (Figure 4G,H). Ropey keratin filaments (Figure 4F, (kf)) could be seen. In contrast to the comparable ambient condition, microbial bodies were rarely visible.

In a high CO_2_ environment, after six weeks no buried feathers were recovered from the CPBB condition (missing data, Figure 4I–L). However, the feathers exposed at the surface (CPBE) are shown in Figure 4M–P. Microbial tunneling (Figure 4M, arrows) was visible in this condition, as was seen in the corresponding ambient condition. Figure 4M is the only sample where original color could be determined; this feather was black. Figure 4N shows a flaky material, possibly representing degraded keratin or, alternatively, fine clay grains. Although Figure 4O reveals aligned keratin fibers (kf), in most samples the outermost surface of keratin was largely degraded. Contrary to the ambient condition, microbial bodies were difficult to see, but their impressions within the degraded keratin were visible as aligned voids (Figure 4M,P; (v)).

When HA was added to pondwater in the elevated CO_2_ environment, preservation was improved in both the buried (CPMB, Figure 4Q–T) and unburied (CPME, Figure 4U–X) conditions. Fine feather structure, including barbs and barbules, remained visible, but a coating of either granular or smooth material (^) could be seen on the buried feathers, that is smoother and less granular than shown in Fig. 3; as a result we used a different symbol. A similar heterogenous material covered the surface of the unburied feathers (Figure 4U–X) to an even greater extent, but beneath this layer, feather barbs appeared intact. Diatoms and other structures were also seen. Geometric mineral crystals (arrowheads) could be seen interspersed with small, round, concave structures approximately 2 µm in diameter (arrows). In Figure 4W, mineral overgrowth on barbules was visible, and thin, plate-like features that could be degraded keratin or clay grains were visible.

Preservation was greatly enhanced in feathers in both buried (CESMB, Figure 4Y–BB) and exposed (CESME, Figure 4CC–FF) conditions in elevated CO_2_ when HA was added. Feather structure was almost perfectly preserved, but interspersed with the barbules in the buried feathers were small plate-like structures. These can be seen more clearly in higher-magnification images (Figure. 4BB). In the unburied feathers (Figure 4CC–FF), preservation was virtually perfect, and no alteration of structure was visible in low-magnification images (Figure 4CC,EE). At higher magnifications, small pockets of microbes appeared on the surface (Figure 4DD,FF, m). Thin, plate-like structures could also be seen.

In the high-CO_2_ environment, feathers buried with E-Pure water alone were almost as well preserved as the preceding condition (CESB, Figure 4GG–JJ). Barbs and barbules were preserved with no evidence of fraying or breakdown, although at higher magnifications, some regions appeared to be covered in a crystalline coating (Figure 4JJ), and rarely, microbodies were seen on the surface (Figure 4HH). In the exposed feathers incubated with E-Pure water only, virtually no degradation was seen (Figure 4KK–NN) and barbs and barbules were intact. Small crystals of mineral precipitate were occasionally seen on the surface of the feather (Figure 4LL, arrowhead), and the presence of a few microbial bodies (m) were also noted on the feather surface. Figure 4MM–NN shows a region where keratin filaments (kf) could be seen extending to, and surrounding, a region of geometric pith (P).

### 3.3. Transmission Electron Microscopy (TEM)

Transmission EM allowed us to study degradation with greater resolution. With ultrathin sections (~90 nm thickness), it is impossible to discern whether black or white regions of the embedded feathers were sectioned. In most cases, where electron-opaque melanosomes were not visualized and no voids were present, we assumed these to be white regions. In all cases, melanosomes, when visible, were most abundant in the feather barbules. They were uniformly opaque to electrons and always embedded in the keratinous matrix, not displayed on the outer surface, as we have previously shown [60]. They did not appear to overlap within the keratin matrix, but were well separated.

Figure 5 shows the various conditions in ambient atmospheres after six weeks of degradation; the first two panels in each row were buried and the second two exposed at the surface. Degradation in pondwater was greater in the buried feathers (PB, Figure 5A,B) than the exposed ones (PE), with a keratinous matrix developing holes. In both buried and exposed feathers, the melanosomes were relatively intact, although some degradation was seen. When *B. licheniformes* were present with pondwater (PBB, PBE; Figure 5E–H), degradation was much more advanced. The keratin matrix was highly degraded in PBB feathers (Figure 5E,F), and the melanosomes showed a great loss of integrity. In the PBE feathers (Figure 5G,H), the keratin matrix showed a loss of smoothness, taking on a “bubbly” texture, but was relatively intact. Melanosomes were not prevalent. This could be because of the region of the feather imaged, as barbules contain more melanosomes than the rest of the feather [61]. When HA was added to the pondwater (PMB, PME; Figure 5I–L), degradation was much less than in previous conditions, and mineral crystals could be seen associated with feather surfaces. Melanosomes and matrix were intact in both cases, although some fraying and loss of integrity was seen in the PME feathers (Figure 5L). Feathers incubated in E-Pure water to which HA was added (ESMB, ESME; Figure 5M–P) showed loss of mineral integrity. No melanosomes were visualized, probably indicating that this was a white feather region. Finally, Figure 5Q–T show feathers incubated in only E-Pure water. Degradation was minimal, but greater than seen when HA was added. Melanosomes were round and almost completely solid; however, in the exposed feathers, some cracking of the keratin matrix was visible.

Figure 6 shows TEM images of feathers degraded in an atmosphere elevated in CO_2_. In most conditions, degradation was advanced compared to that seen in ambient conditions. Figure 6A–B, depicting CPB buried feathers, show advanced degradation, with many melanosomes completely degraded, and those remaining showing loss of integrity. Exposed feathers (CPE) fared better, with elongated and round melanosomes present within the keratin matrix. When *B. licheniformis* was added to the pondwater, all feathers were completely degraded at six weeks, so as above, images (Figure E–H) were taken after four weeks. The buried feathers (CPBB) showed greater degradation, with most melanosomes no longer intact. In the exposed feathers (CPBE; Figure 6G,H), voids can also be seen where melanosomes once resided. In high-CO_2_ environments, when HA was added to pondwater, degradation was greatly decreased. Melanosomes showed slight degradation in the buried condition (CPMB; Figure 6I,J), with some apparently lysed. Electron-opaque, needle-like mineral deposits (mc) were seen outside of the keratin matrix. Exposed feathers (CPME; Figure 6K–L) showed better preservation, with little degradation of either melanosomes or keratin. There was an unidentified growth on the outer surface of the keratin that was most likely organic, based on its electron translucent character.

Buried feathers incubated in elevated CO_2_ with E-Pure water to which HA had been added (CESMB; Figure 6M,N) showed relatively advanced degradation of melanosomes, leaving voids, but with little degradation of the keratin matrix in which they were embedded. Exposed feathers (CESME; Figure 6O,P) showed similar degradation. The keratin matrix was relatively unaltered. Melanosomes were rare, but empty voids were present, supporting the idea that these feathers contained pigment organelles.

Feathers incubated in E-Pure water only showed little overall degradation. Buried feathers (CESB; Figure 6Q,R) showed little keratin degradation but no melanosomes or voids, and therefore probably represent white feather regions. Undegraded, hollow melanosomes were visible embedded deep within the keratin matrix in the exposed condition (CESE; Figure 6S,T).

### 3.4. In Situ Immunohistochemistry (IHC)

We tested the hypothesis that degradation influences antibody binding. Figure 7 shows the response of feathers in different degradation conditions when exposed to a polyclonal antiserum raised against feathers [62]. All images were taken under identical parameters. In the ambient condition, no reduction in intensity of binding was seen, even in the feathers showing the greatest morphological damage (Figure 7A,B,E,F), consistent with preservation of epitopes after structural integrity was lost. Antibody specificity is supported by the lack of binding to the embedding material, or to regions where tissues were completely missing. Negative controls of no primary antibody applied, but all other steps being the same (not shown), indicates that the signal did not arise from non-specific binding of the secondary antibody or fluorescent label. Melanized feather regions (Figure 7A,C,I,K) were more easily visualized in transmitted LM than those regions apparently lacking melanin (Figure 7E,G,M,O,Q,S), but antibodies bound with equal intensity with both types degraded for the duration of this experiment.

When feathers were degraded under the same conditions as in Figure 7, but in elevated CO_2_ atmospheres, the results were different. Buried feathers exposed to pondwater only (CPB; Figure 8A,B) showed a highly degraded area covering a more intact region. Polyclonal antibodies bound with high avidity, but no melanized regions could be seen with certainty in the transmitted LM (Figure 8A). The exposed feathers (CPE) with pondwater demonstrated better microscopic integrity. Regions of melanized barbules (Figure 8C) could be seen in LM and antibodies bound with the same level of intensity (Figure 8D), as seen in Figure 7. When feathers were buried and incubated with pondwater and *B.licheniformis* in elevated CO_2_ (CPBB), no feathers remained for analyses; data are shown for this condition in feathers after four weeks degradation (Figure 8E,F). Integrity was greatly reduced, despite melanin in the visible barbs (Figure 8E). Surface-exposed feathers (CPBE; Figure 8G,H) showed minimal degradation, despite containing less or no melanin. Figure 8I–L shows buried (Figure 8I,J) and exposed (Figure 8K,L) feathers incubated with pondwater with added HA. Both buried and exposed feathers showed melanized regions that remained virtually intact, with no visible degradation, and both bound antibodies with equal and intense avidity. Similarly, Figure 8M–P shows buried (M,N) and exposed (O,P) feathers in E-Pure water to which HA had been added. Virtually no degradation of epitopes was seen, and no substantial differences between lightly melanized and buried (M,N) or unmelanized and exposed (O,P) were seen. Non-melanized buried (Figure 8Q,R) and melanized, exposed (Figure 8S,T) feathers in E-Pure water only were comparable, showing no tissue degradation and no reduction in antibody binding under these conditions.

## 4. Discussion

We showed, using multiple methods, that the degradation of organics is qualitatively and/or quantitatively different in atmospheres elevated to Mesozoic CO_2_ levels (or higher) than what is observed in ambient atmospheres. Some microbes are known to precipitate CO_2_ as carbonate minerals [63,64,65], and we sought to determine whether this precipitation could be increased in atmospheres elevated in CO_2_. We chose hydroxyapatite because microbes are known to precipitate this mineral [66], and because the solubility of HA rises with increased CO_2_ [67]. Because it has been shown that organic molecules can be preserved preferentially in intracrystalline regions of bone [68], we predicted that we may see better molecular preservation in extant samples subjected to elevated CO_2_ through rapid stabilization by mineral precipitation. We used both melanized and non-melanized feathers to test the effect of pigment on preservation [47,69]. We added feather-degrading microbes to natural microbial biomass to see whether the action of these organisms was increased in high CO_2_ atmospheres, and we tested the effect that increasing the concentration of HA would have on preservation and degradation in varying atmospheres. Finally, we compared preservation when feathers were exposed to waters containing a natural microbial population vs. E-Pure water.

In ambient atmospheres, degradation was greatest when feather-degrading bacteria were added to feathers in the natural microbiota of pondwaters. In general, buried feathers were more highly degraded than surface-exposed feathers, and black regions retained more structural integrity than did white feather regions. The greater degradation observed for buried feathers was not predicted, but we hypothesize that burial gave greater access to microbes by keeping the feathers uniformly damp. In the other conditions, there was little difference, but importantly, adding HA to the waters containing microbes greatly increased preservation, perhaps by slowing the microbially mediated degradation undertaken by microbes naturally present in feathers. In E-Pure water, with or without minerals, virtually no degradation was seen, and black feathers did not differ from white in preservation, supporting a specific role for microbially mediated precipitation in preservation. Feathers in elevated CO_2_ showed much greater degradation, whether buried or exposed, in both natural pondwaters and with added microbes; however, in the presence of added HA, degradation was essentially halted, and preservation was relatively greater in elevated CO_2_. SEM data for both natural waters and waters with added microbes demonstrated multiple holes and tunneling not seen in ambient atmospheres, suggesting upregulation of microbial enzymes of degradation in elevated CO_2_. Increased microbial activity in response to elevated CO_2_ was noted in other studies as well [70,71,72,73]. The increase in preservation in both atmospheric conditions when HA was added suggests the deposition of mineral-arrested degradation in both buried and unburied states, as suggested elsewhere [74,75], either by adsorbing enzymes of degradation and inactivating them (e.g., [76], by encasing organics, thus making them inaccessible [77]), or both, but this process was relatively enhanced in elevated CO_2_. Where color could be detected, there was no measurable difference in preservation/degradation in most conditions. Keratin filaments and intact barbs, barbules, and sheaths showed little to no degradation under the elevated CO_2_ + mineral condition. Fungal hyphae, prevalent in the ambient conditions, were not discerned in elevated CO_2_ except in the pondwater-only condition.

TEM illustrated high levels of degradation of the keratin sheath in pondwater conditions, both buried and unburied, but melanosomes remained relatively intact, though some degradation of these bodies was visible. However, in buried feathers with added *B. licheniformis*, melanosome and keratin degradation were both advanced over other conditions. Crystal deposits were seen when HA was added, and in these conditions, the feathers appeared unaltered. Patterns were similar in the elevated CO_2_ atmospheres for both pondwater and with added microbes, again with datapoints representing only four weeks in the buried condition. In elevated CO_2_ with added HA, melanosomes appeared more degraded than did keratin, leaving many voids in the buried feathers.

In ambient atmosphere, antibody binding was intense in all cases, with little difference noted between all conditions shown here. Burial state, added microbes, and presence or absence of melanin did not appear to affect the recognition of keratin epitopes by these polyclonal antibodies, and binding occurred even when structural integrity was not well maintained. Although no material persisted to the six-week timepoint in buried feathers with added *B. licheniformis* in elevated CO_2_, the only case where antibody binding was detectably reduced was in the buried feathers after four weeks. Both structural integrity and epitope recognition were greatly diminished.

Processes resulting in the preservation of normally labile and easily degraded organic materials have been of significant interest to the paleontological community, because examples of hair, skin, feathers, and internal organs do not follow standard models of fossilization. It is significant that incidences of exceptional soft tissue preservation are not evenly distributed through time, but occur more frequently in pre-Cenozoic deposits [78,79] during time periods when atmospheric CO_2_ was elevated over today’s levels [80,81]. In addition, Cenozoic lagerstätte are few, but generally correlate to short periods of elevated CO_2_ [82,83,84]. We propose that this uneven distribution of exceptionally preserved deposits may be due in part to the response of microbes to elevations in atmospheric CO_2_ and the subsequent acidification of pore waters, making mineral ions more available for precipitation. This preliminary study demonstrates the need for further investigation into the role of both microbes and CO_2_ in preserving organic remains in fossils.

## 5. Conclusions

Degradation by all measures was greatest in buried feathers in elevated CO_2_. However, the addition of HA slowed degradation, whether with natural pondwaters or E-Pure water, and this effect was greater, relative to level of degradation without minerals, in elevated CO_2_. The rapid precipitation of minerals on organics outpaced decay in these conditions, illustrating a possible role in exceptional preservation, as suggested by [85,86,87,88] and others. Similarly, antibody recognition as a proxy for molecular preservation was still present, even in feathers showing microstructural degradation. The implications for fossilization support the hypothesis that degradation proceeded differently in the elevated CO_2_ atmospheres of the Mesozoic. Although degradation was more rapid and more complete in buried feathers, antibody binding was not greatly affecting during the time interval of this experiment, suggesting a lack of direct correlation between histological integrity and antibody binding. In most cases, keratin preservation was equal to or greater than melanosomes under the conditions of this experiment, whatever the atmospheric conditions. We tested the role of apatite because, in atmospheres elevated in CO_2_, as in the Mesozoic, warmer waters and the acidification of pore waters would allow an increase in the concentration of this solubilized mineral. Experimental taphonomy designed to address questions of degradation in the past, particularly in deep time intervals, will be misleading unless atmospheric composition and the effects of greenhouse gases on degrading microbes are taken into account.

## Figures and Tables

**Figure 1 biology-11-00703-f001:**
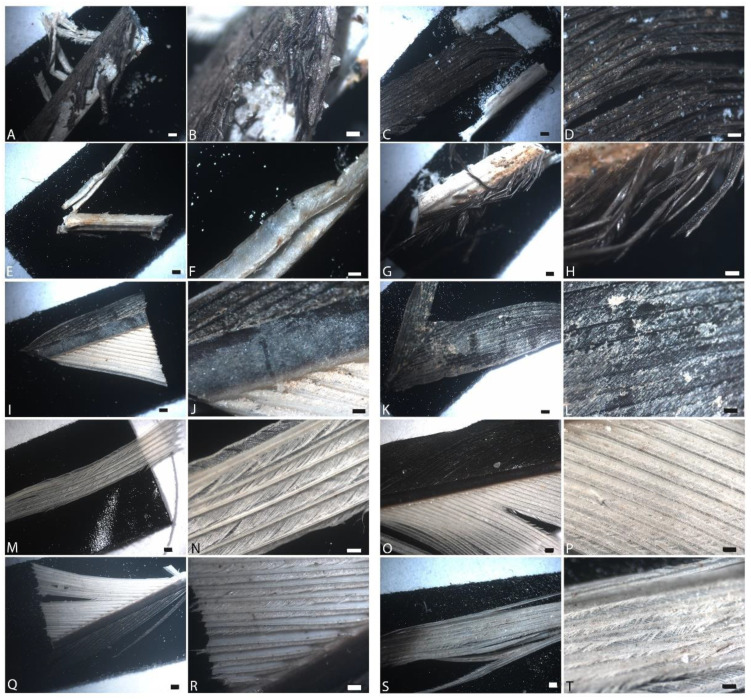
Feathers degraded in ambient atmospheres after 6 weeks. (**A**,**B**) Buried and (**C**,**D**) exposed in natural pondwater; (**E**,**F**) buried and (**G**,**H**) exposed feathers with pondwater and added microbes; (**I**,**J**) buried and (**K**,**L**) exposed feathers with pondwater and added HA; (**M**,**N**) buried and (**O**,**P**) exposed feathers with E-Pure water and HA; (**Q**,**R**) buried and (**S**,**T**) exposed feathers with E-Pure water only. Scale is 500 µm for the 1st and 3rd columns, and 200 µm for the 2nd and 4th columns.

**Figure 2 biology-11-00703-f002:**
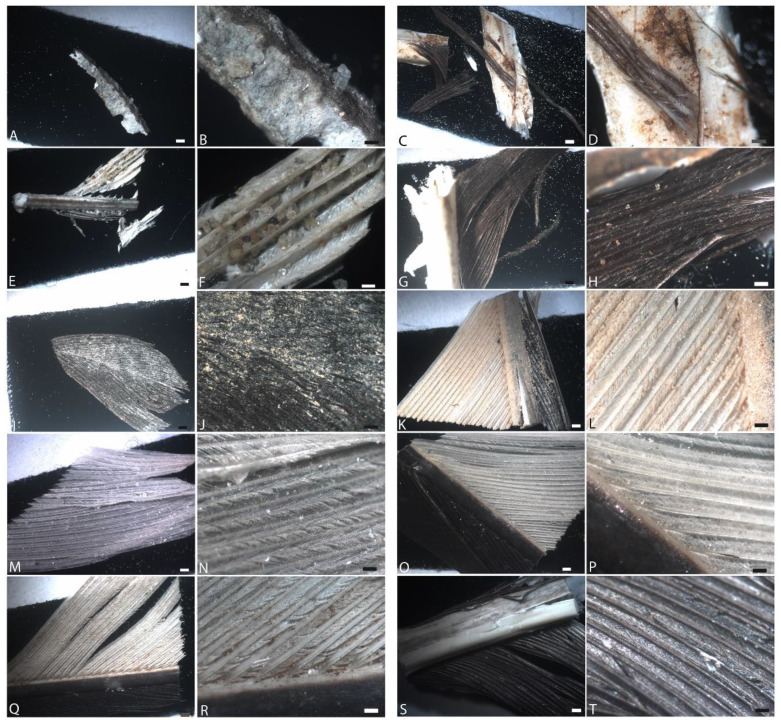
Feathers degraded in elevated CO_2_ atmospheres after 6 weeks. (**A**,**B**) Buried and (**C**,**D**) exposed with natural pondwater; (**E**,**F**) buried (4 weeks) and (**G**,**H**) exposed feathers with pondwater and *B. licheniformis*; (**I**,**J**) buried and (**K**,**L**) exposed feathers with pondwater and added HA; (**M**,**N**) buried and (**O**,**P**) exposed feathers with E-Pure water and HA; (**Q**,**R**) buried and (**S**,**T**) exposed feathers with E-Pure water only. Scale bar for 1st and 3rd columns is 500 µm, and for the 2nd and 4th, 200 µm.

**Figure 3 biology-11-00703-f003:**
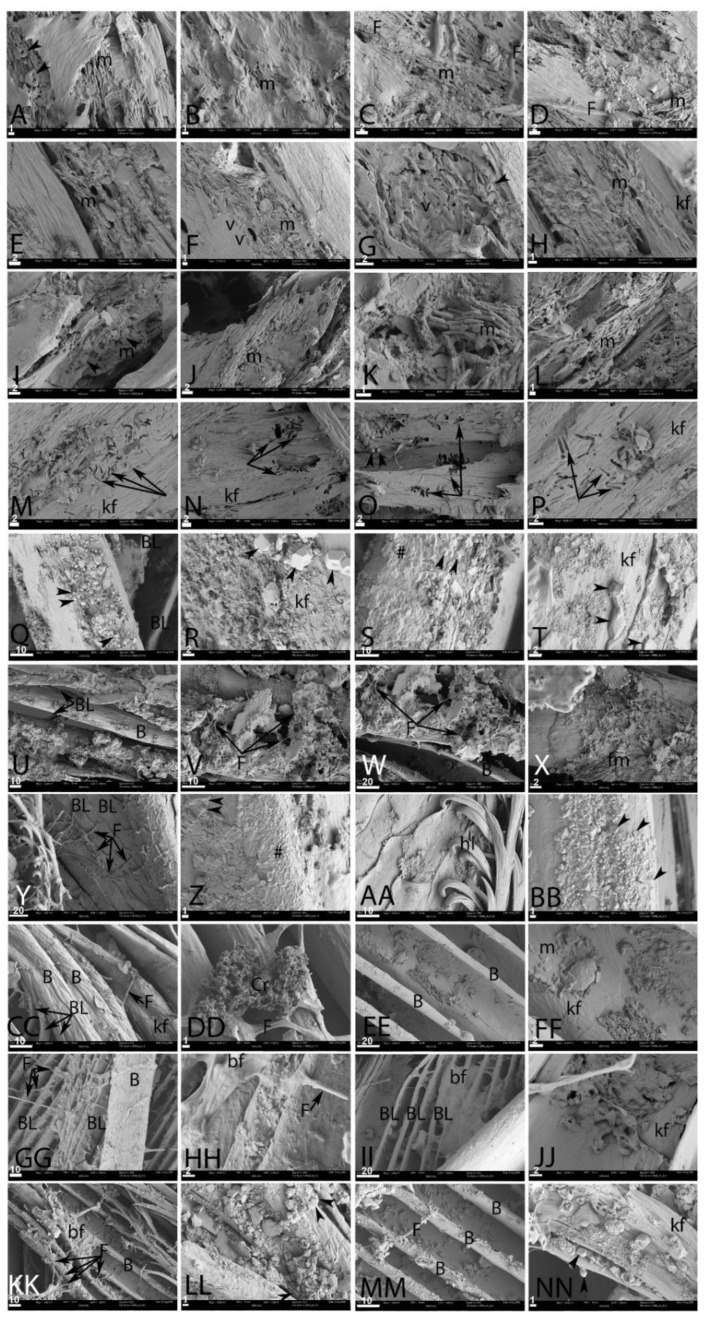
Feathers degraded under ambient conditions. (**A**–**D**) Buried in pondwater; (**A**,**B**) are black and (**C**,**D**) are white. (**E**–**H**) are feathers exposed at the surface; colors are not discernable. (**I**–**L**) Feathers buried in pondwater to which *B. licheniformis* has been added. (**M**–**P**) Surface-exposed feathers in pondwater with *B. licheniformis*; color is not discernible in (**I**–**P**). Arrows show microbial tunneling. (**Q**–**T**) represent feathers buried in pondwater with added HA; (**Q**,**R**) are black and (**S**,**T**) are white. A film (#) covers some regions; also (**U**–**X**) are exposed feathers with pondwater and HA; colors are not discernible. (**Y**–**BB**) are feathers buried with E-Pure water and HA; (**Y**,**Z**) are black and (**AA**,**BB**) are white. (**CC**–**FF**) are exposed feathers in E-Pure water with added HA; (**CC**,**DD**) are black and (**EE**,**FF**) are white. Finally, (**GG**–**JJ**) are buried and (**KK**–**NN**) are exposed in E-Pure water only; (**GG**,**HH**,**KK**,**LL**) are black and (**II**,**JJ**,**MM**,**NN**) are white. Scale as indicated in μm. kf, keratin filaments; bf, biofilm; m, microbial bodies; v, voids; F, fungal hyphae.

**Figure 4 biology-11-00703-f004:**
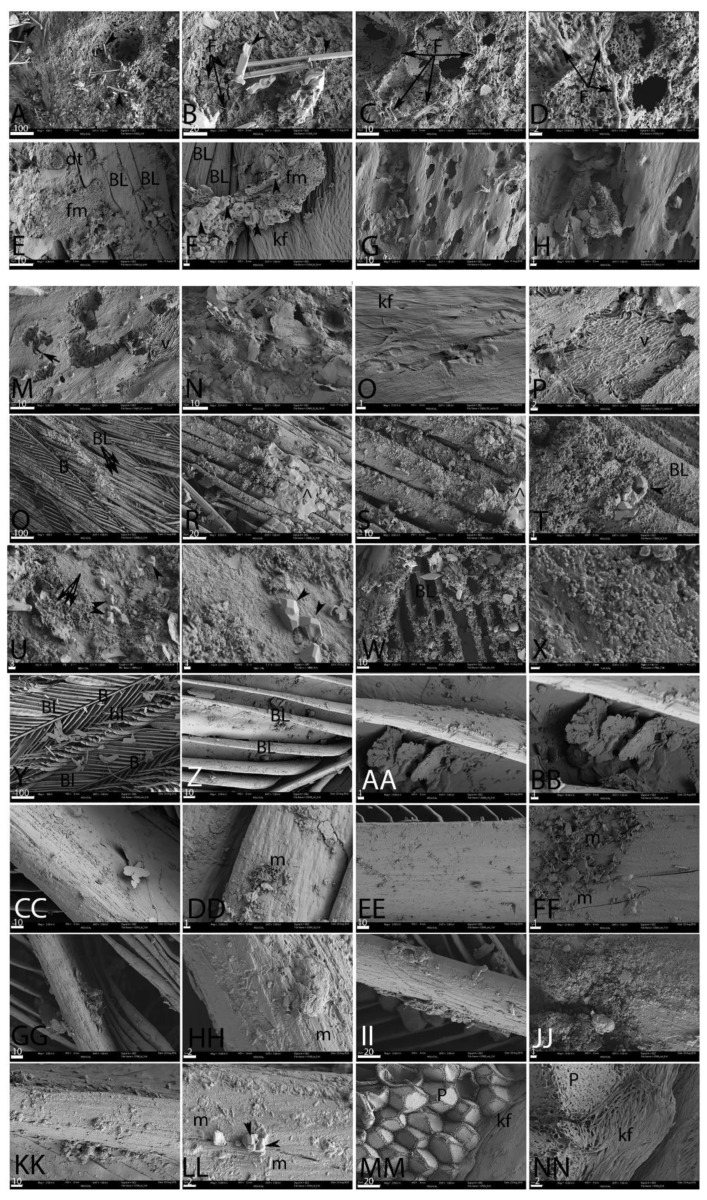
Feathers degraded for 6 weeks under elevated CO_2_. 4 (**A**–**D**) are buried in pondwater, color is not discernible. (**E**–**H**) are surface-exposed; (**E**,**F**) are black and (**G**,**H**) are white. Feathers buried in pondwater to which *B. licheniformis* was added did not persist to 6 weeks and no data are shown. (**M**–**P**) Surface-exposed feathers in pondwater with *B. licheniformis*. Color is only discernible in (**N**), which is black. (**Q**–**T**) represent feathers buried in pondwater with added HA; all feathers shown are black. (**U**–**X**) are surface-exposed with pondwater and HA; (**U**,**V**) are black and (**W**,**X**) are white. (**Y**–**BB**) are feathers buried with E-Pure water and HA; all are white. (**CC**–**FF**) are exposed feathers in E-Pure water with added HA; (**CC**,**DD**) are black and (**EE**,**FF**) are white. (**GG**–**NN**) Incubated in E-Pure water only; (**GG**–**JJ**) are buried and (**KK**–**NN**) are surface-exposed. (**GG**,**HH**,**KK**,**LL**) are black, (**II**,**JJ**) are white, and (**MM**,**NN**) are indeterminate. Scales as indicated, in μm. kf, keratin filaments; bf, biofilm; m, microbial bodies; v, voids; F, fungal hyphae; ^, amorphous film.

**Figure 5 biology-11-00703-f005:**
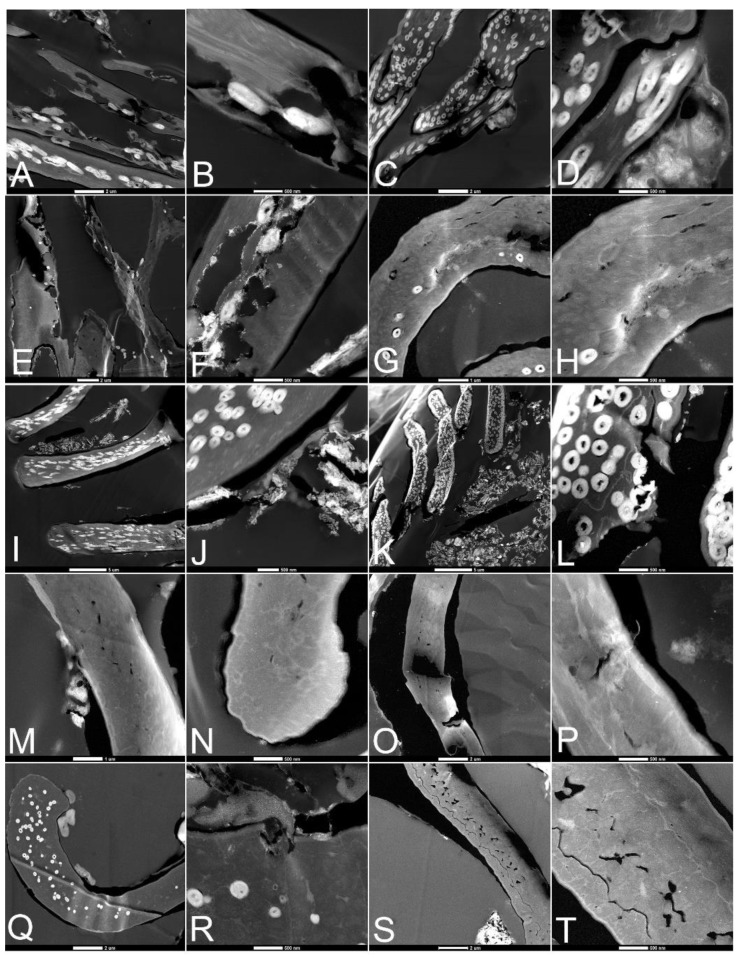
Feathers degraded in ambient conditions. (**A**–**D**) Exposed to pondwater; (**A**,**B**) buried and (**C**,**D**) exposed. (**E**–**H**) Pondwater with *B. licheniformis* added; (**E**,**F**) buried and (**G**,**H**) exposed. (**I**,**L**) are feathers incubated with pondwater and added HA; (**I**,**J**) buried and (**K**,**L**) exposed. Electron-opaque, needle-like mineral crystal (mc) deposits are seen exterior to the keratin of the feather barbules. (**M**–**P**) show feathers incubated in E-Pure water to which HA has been added. No melanosomes are seen, probably indicating that these are white feather regions, although early degradation is a less likely explanation. (**Q**–**T**) show feathers incubated in E-Pure water only. (**S**) low magnification, (**T**) higher magnification showing some cracking of the feather matrix. Scale bars as indicated.

**Figure 6 biology-11-00703-f006:**
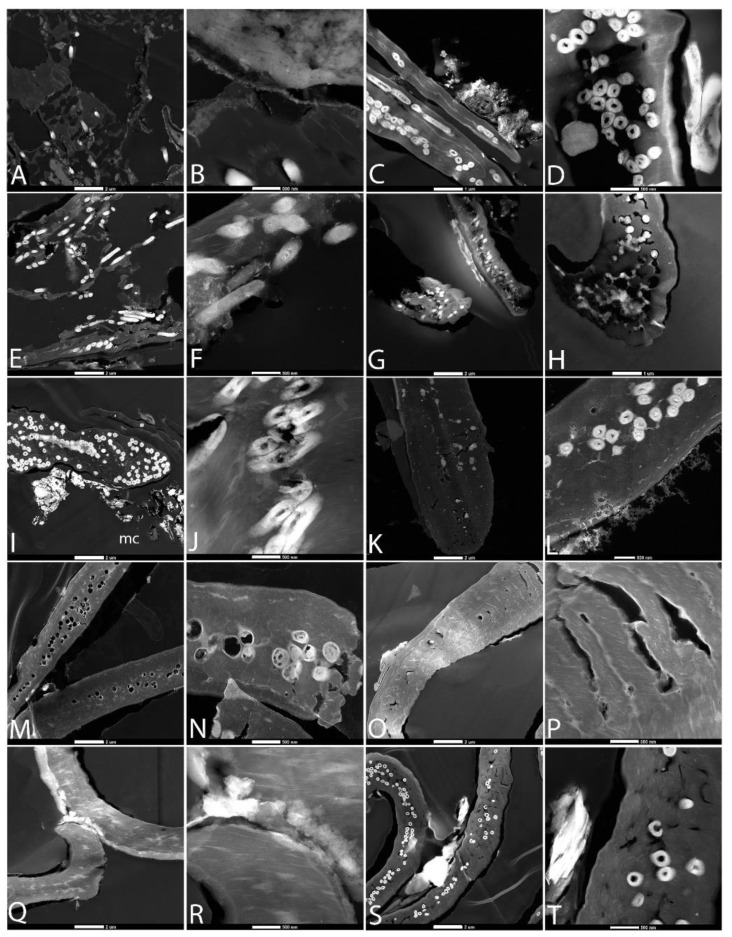
Transmission electron microscopy (TEM) of feathers degraded in elevated CO_2_ atmosphere. (**A**,**B**) are buried and (**C**,**D**) are exposed in natural pondwaters. (**E**–**H**) show feathers in natural pondwater with *B. licheniformis* added; (**E**,**F**) are buried (after 4 weeks) and (**G**,**H**) are exposed. (**I**–**L**) Pondwater to which HA has been added; (**I**,**J**) are buried and (**K**,**L**) are exposed at the surface. (**M**,**N**) are buried and (**O**,**P**) are exposed feathers degraded in E-Pure water to which HA had been added. Finally, (**Q**–**T**) are feathers in elevated CO_2_, degraded in E-Pure water only; (**Q**,**R**) are buried and (**S**,**T**) are exposed. Scale bars as indicated.

**Figure 7 biology-11-00703-f007:**
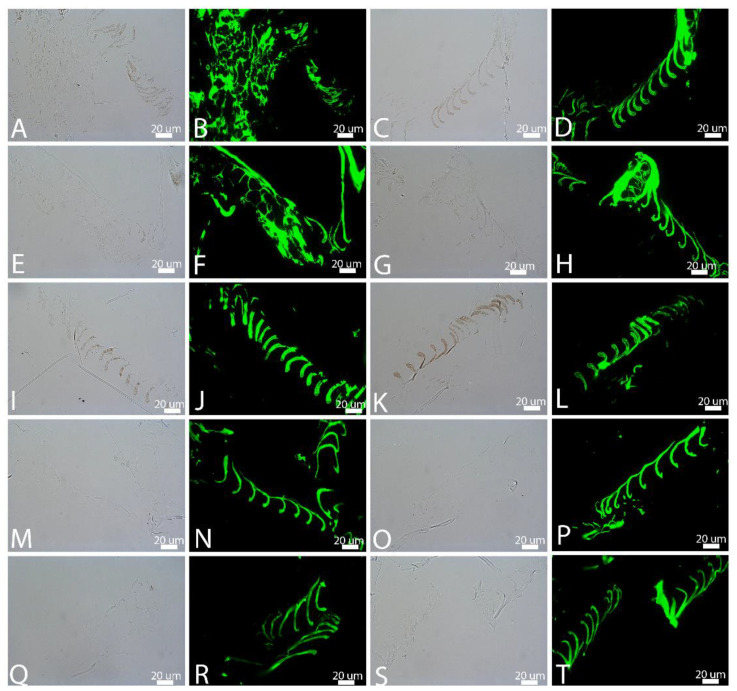
In situ immunochemistry of feathers degraded in ambient atmosphere, then exposed to a polyclonal antiserum raised against whole feather extract and visualized in transmitted light (1st and 3rd image of each row) or using an FITC filter (2nd and 4th image). (**A**,**B**) are buried and (**C**,**D**) are exposed in natural pondwaters. (**E**–**H**) show feathers in natural pondwater with added *B. licheniformis* at 4 weeks; (**E**,**F**) are buried and (**G**,**H**) are exposed. (**I**–**L**) Pondwater to which HA has been added; (**I**,**J**) are buried and (**K**,**L**) are exposed at the surface. (**M**,**N**) are buried and (**O**,**P**) are exposed feathers degraded in E-Pure water to which HA has been added. Finally, (**Q**–**T**) are feathers degraded in E-Pure water only; (**Q**,**R**) are buried and (**S**,**T**) are surface-exposed. Transmitted light images are modified to increase contrast; see Appendix A.

**Figure 8 biology-11-00703-f008:**
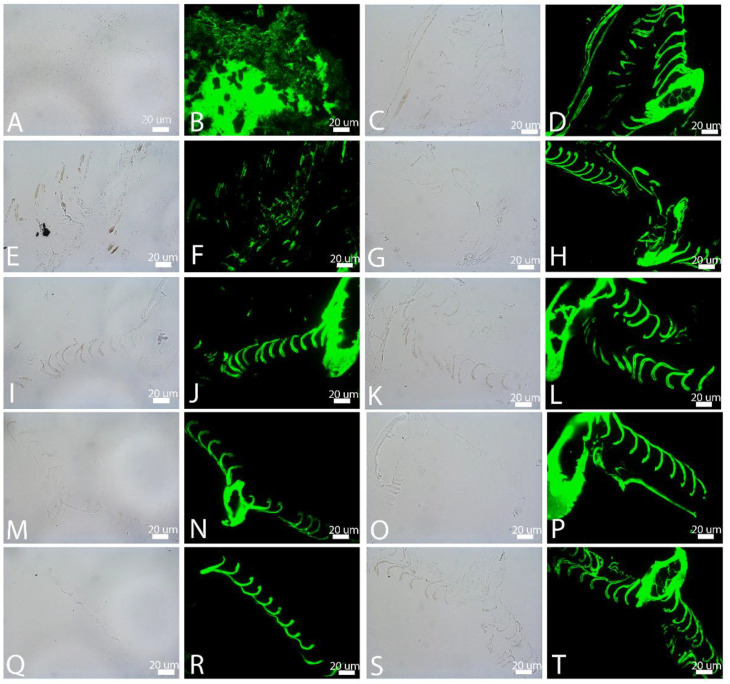
In situ immunochemistry of feathers degraded in elevated CO_2_ atmosphere, then exposed to a polyclonal antiserum raised against whole feather extract and visualized in transmitted light (1st and 3rd image of each row) or using an FITC filter (2nd and 4th image). (**A**,**B**) are buried and (**C**,**D**) are exposed in natural pondwaters. (**E**–**H**) show feathers in natural pondwater with added *B. licheniformis*; (**E**,**F**) are buried (after 4 weeks) and (**G**,**H**) are exposed. (**I**–**L**) Pondwater to which CaPO_4_ has been added; (**I**,**J**) are buried and (**K**,**L**) are exposed at the surface. (**M**,**N**) are buried and (**O**,**P**) are exposed feathers degraded in E-Pure water to which HA had been added. Finally, (**Q**–**T**) are feathers in elevated CO_2_, degraded in E-Pure water only; (**Q**,**R**) are buried and (**S**,**T**) are surface-exposed.

**Table 1 biology-11-00703-t001:** Experimental conditions to test their effect on preservation and/or degradation.

Ambient Atmosphere	Elevated CO_2_ Atmosphere
Pondwater, buried (PB)	Pondwater, buried (CPB)
Pondwater, exposed (PE)	Pondwater, exposed (CPE)
Pondwater + *Bacillus licheniformis*, buried (PBB)	Pondwater + *B.licheniformis*, buried (CPBB)
Pondwater + *B.licheniformis*, exposed (PBE)	Pondwater + *B.licheniformis*, exposed (CPBE)
Pondwater + Hydroxylapatite, buried (PMB)	Pondwater + HA, buried (CPMB)
Pondwater + HA, exposed (PME)	Pondwater + HA, exposed (CPME)
E-Pure water, sand + HA, buried (ESMB)	E-Pure water, sand + HA, buried (CESMB)
E-Pure water, sand + HA, exposed (ESME)	E-Pure water, sand + HA, exposed (CESME)
E-Pure water, sand, buried (ESB)	E-Pure water, sand, buried (CESB)
E-Pure water, sand, exposed (ESE)	E-Pure water, sand, exposed (CESE)

## Data Availability

All data dealing with this study are reported in the paper.

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
