# Peer review of "Environmental Factors Affecting Feather Taphonomy"

_biology, 2022, doi:10.3390/biology11050703_

Round 1

Reviewer 1 Report

There is a lot of interesting data in your manuscript. Unfortunately, that data is poorly organised and presented. It is difficult to read and even more difficult to digest. It is also hard to confirm or evaluate your interpretations because the images you present are of too low resolution. Not even the Supplementary data file contains high resolution images.

Too much information is squeezed into your figures. Consequently, they are hard to follow and difficult to match up with descriptions in the text.

The following issues must be addressed and the manuscript resubmitted.

1) The aims of the research and the hypotheses tested must be described very concisely in either the Introduction or the Methods section. As it stands the reasons for doing the experiments are scattered through the manuscript. Please make these very obvious - and if necessary itemised - right at the beginning.

2) Please provide much more details in the methods section. It is not clear how your samples were degraded. How deep was the water, how deep was the sediment, at what depth were the samples buried, and what were the sizes of the sample chambers? (Perhaps a diagram would suffice here) How much atmosphere was above the water? Were the samples degraded in the dark or was there a day/night cycle in illumination? What was ambient temperature? I take it no controls/blanks were  run but then it would be difficult to know how to do that usefully.

3) How were samples retrieved? How were they dried? Air dried or freeze dried? What effects would drying regimes possibly have? The mineral formation on the surfaces of the degraded feathers may well have been drying artefacts. It is impossible to tell from the data provided.

4) More detail is required about spiking the water with calcium phosphate solutions. A reference [62] is cited in the text but there is no reference 64 in the References. And why 6mMol/litre? (although the parent solutions are 0.1 Mol/litre) That exceeds the solubility of HAP at the likely pH of your solutions. What will the influence of elevated carbonate in your solutions likely to be on the solubility of HAP and the sizes of the crystals that if forms?

5) Table 1 is actually not very useful when reading the text. Can I suggest you assign each experimental regime an alpha-numeric code that is easy to interpret/remember and that these codes are used in the text below? Less wordy and more elegant. 

6) Divide the figures into ones with fewer images that cover just two of the experimental regimes (normal air/elevated CO2) or sediment surface/ within sediment. Then the figure legends can be shorter and there will be less need for AA, BB, etc.

7) The scales are completely inadequate! Do not rely on the scales automatically added by the SEM/TEM. They are illegible when the image is the size of a postage stamp. Get rid of the data bar at the foot of electron micrographs. They are illegible anyway and waste space. Add proper scales that help to interpret the images.

8) Take more care with the terminology you use and stick to it (I know that more than one person is writing this paper but try to be self consistent). The term "incubating" appears on line 119. This term is not used in the Methods section. The various experimental conditions are complex enough without adding further obfuscating terms.

9) Please be more clear about spiking the experimental environments with bacteria. In the Methods we have "or addition of feather degrading microbes (B. licheniformis) to existing microbial populations in pondwater" which implies these were added to the water. Then on line 180 we have "When feathers were soaked in pure cultures of B. licheniformis then added to the pondwater flora". One might expect different outcomes from these two approaches.

10) The authors describe various crystal or different morphology forming on or around their samples. But go on to say that these were also visible with samples that were not spiked with HAP. Considering the authors used SEM and TEM it is disappointing that no attempt was made to identify these crystals (EDX or electron diffraction). Because it is impossible to see scales in their micrographs it is not even possible to approximate their size and no attempt is made in the text to give their dimensions.

11) Figure 7 has the potential to be spectacular. However, the impact is mostly lost because the transmission light micrographs are practically illegible. Can something not be done to enhance the contrast (describing how and why this was done in the text)?

Author Response

Please see attached document.  Reviewer comments are listed and our direct response to this is given in italics.

Reviewer 2 Report

Line 19: From an ichthyological point of view this can indeed be questioned when you think of the complexity of scales.

Line 23-24: This is not correct, theropods and birds are not sister-groups, modern birds (Aves) and extinct birds are all deeply nested within theropods. Birds are theropods.

Line 31-32: The single feather is not the holotype specimen anymore.

Methods: The physico-chemical properties of the “pond water” and the used sediment (pond sediment and sand) are not given. You will need to add sedimentological data (mineralogy, grain size, porosity, etc.). Also, the use of what I assume is quartz sand is untypical for feather preservation in the fossil record. Feather are usually either preserved in fine-grained limestone (e.g., Solnhofen) or in siliciclastic mudstones, all very fine grained sediments.

CO2 levels surely varied throughout the Mesozoic and were still very high during the Paleogene. Please give the time frame (Late Jurassic?) you want to emulate, and precisely state why you chose the applied CO2 level and on which reconstruction (modelling, proxies) this is based, this is far from trivial. I am also highly skeptical of the increased acidification of freshwater during the Mesozoic that you presume, as today, pH of the water would primarily be the result of watershed bedrock mineralogy and local hydrology.

Figures 3&4: The individual subfigures are too small. Lettering and arrows are sometimes not properly readable.

Discussion: The discussion lacks a comparison of the results with the fossil record, e.g. Jehol Biota, Solnhofen, etc. from which feather preservation is known.

Can these conclusions be confirmed from the fossil record? Is there a difference between feather preservation in Neogene sediments (low CO2) and Mesozoic-Paleogene sediments? I do not think these experiments demonstrate a large effect of CO2 under natural conditions: 1) mud and mudstones, where feathers are usually preserved, would enclose feathers from porewaters, 2) feather preservation is also more common in anoxic conditions, which was not accounted for, 3) a higher acidification of pore waters would very much depend on the local sediment mineralogy and not mainly on atmospheric CO2.

Author Response

please see attached document, again, reviewer comments are shown, and our response to them is in italics.

Reviewer 3 Report

Dear authors

This is well written, interesting and important paper on the Taphonomy of feathers.  All conclusions are supported by the data.  I can only agree with the authors. I just recommend a further paragraph in the discussion section to situate your findings in comparison with other similar studies. I have indicated few papers, but you can feel free to compare with any other study.
In the attached files you can see minor suggestions.

Kind regards

Author Response

Reviewer comments were given on the review pdf. we responded by changing what we read, and address a few of the comments in the attached document. otherwise we used track changes on the document to show our responses.

Round 2

Reviewer 1 Report

1) Most of the comments are simple typos or suggestions for language/style.

2) The manuscript is much better. The authors did a good job.

3) Some comments are only for the interest of the authors. They need not add/change anything unless they want to. There are quite a few (but rather old) publications on buried hair (either in forensic contexts or describing archaeological woolen textiles) that show similar kinds of deterioration as the authors illustrate (Wilson et al x2 - citations added in comments). They could add a reference if they can track down the papers and read them.

4) Figures are spectacular. I'm glad I could see them properly.

5) A couple of minor things I really would like the authors to fix before its published. a) The idea of the methods section is so other scientists can repeat the experiments. They will need to know about temperature and illumination. b) The authors did not address the matter of HAP and elevated carbonate levels in the water. Rapid reservation of soft tissues (e.g. muscle) by phosphatisation, involving HAP, is well known in palaeontology. If the authors add HAP to the mix the potential for this is obvious. The authors give no rational for adding HAP in their experiments (why not elevated Fe, Mn or Mg?).

Furthermore, at elevated CO2 levels the HAP is more soluble and when it does precipitate it forms much smaller, void-filling crystals. A layer of small carbonated HAP crystals might be better at protecting/stabilising degrading organics than larger HAP crystals (that might form in lower CO2 levels). I strongly suggest the authors make a suggestion this may be the case. 6) On the subject of HAP, why do the authors use the term "mineral salts" occasionally in the text? I assume they refer to the added Ca and P but why add confusion? 7)

I would like to see the authors add some suggestions of further research directions that could be followed, raised by interesting unresolved questions in their research.
